# A single Gal4-like transcription factor activates the Crabtree effect in *Komagataella phaffii*

Özge Ata[1,2], Corinna Rebnegger[1,3], Nadine E. Tatto[1,4,5], Minoska Valli[1,4], Teresa Mairinger[4,6,8], Stephan Hann[4,6], Matthias G. Steiger [1,4], Pınar Çalık [2,7] & Diethard Mattanovich [1,4]

The Crabtree phenotype defines whether a yeast can perform simultaneous respiration and fermentation under aerobic conditions at high growth rates. It provides Crabtree positive yeasts an evolutionary advantage of consuming glucose faster and producing ethanol to outcompete other microorganisms in sugar rich environments. While a number of genetic events are associated with the emergence of the Crabtree effect, its evolution remains unresolved. Here we show that overexpression of a single Gal4-like transcription factor is sufficient to convert Crabtree-negative *Komagataella phaffii* (*Pichia pastoris*) into a Crabtree positive yeast. Upregulation of the glycolytic genes and a significant increase in glucose uptake rate due to the overexpression of the Gal4-like transcription factor leads to an overflow metabolism, triggering both short-term and long-term Crabtree phenotypes. This indicates that a single genetic perturbation leading to overexpression of one gene may have been sufficient as the first molecular event towards respiro-fermentative metabolism in the course of yeast evolution.

---

[1] Department of Biotechnology, University of Natural Resources and Life Sciences, 1190 Vienna, Austria. [2] Department of Biotechnology, Graduate School of Natural and Applied Sciences, Middle East Technical University, 06800 Ankara, Turkey. [3] CD-Laboratory for Growth-Decoupled Protein Production in Yeast, Department of Biotechnology, University of Natural Resources and Life Sciences, 1190 Vienna, Austria. [4] Austrian Centre of Industrial Biotechnology (ACIB), 1190 Vienna, Austria. [5] School of Bioengineering, University of Applied Sciences FH-Campus, 1190 Vienna, Austria. [6] Department of Chemistry, University of Natural Resources and Life Sciences, 1190 Vienna, Austria. [7] Department of Chemical Engineering, Industrial Biotechnology and Metabolic Engineering Laboratory, Middle East Technical University, 06800 Ankara, Turkey. [8] Present address: Swiss Federal Institute of Aquatic Science and Technology (EAWAG), 8600 Dübendorf, Switzerland. Correspondence and requests for materials should be addressed to D.M. (email: diethard.mattanovich@boku.ac.at)

The fermentative activity of yeast has been used to make bread and fermented beverages by mankind since its early settlement. Thus the physiological capabilities of yeast are directly linked to human development and culture. These processes depend fundamentally on the respiro-fermentative phenotype also called the Crabtree effect of some yeast species, most prominently baker's yeast (*Saccharomyces cerevisiae*).

A similar phenomenon as the Crabtree effect can be observed in cancer cells which have high glycolytic rates and lactic acid fermentation, termed the Warburg effect[1]. The Crabtree effect of *S. cerevisiae* is seen as a unicellular model to study the genetic and metabolic control of the Warburg effect[2–6].

Being a fundamental feature of yeast metabolism, the Crabtree phenotype defines whether a yeast can perform simultaneous respiration and fermentation under aerobic conditions at high growth rates. Crabtree-positive (respiro-fermenting) yeasts, such as *S. cerevisiae*, *Vanderwaltozyma polyspora*, *Torulaspora franciscae*, *Lachancea waltii*, and *Lachancea kluyveri* (syn. *Saccharomyces kluyveri*) can exhibit alcoholic fermentation in the presence of oxygen until glucose reaches low concentrations[7,8]. Nevertheless, rather few yeasts show this phenotype. In most cases, glucose uptake and glycolysis are strictly regulated to match the TCA cycle capacity, so the exclusive use of respiration leads to full oxidation of glucose to $CO_2$ and water with maximum energy yield under aerobic conditions. These yeasts, such as *Yarrowia lipolytica*, *Candida albicans*, *Eremothecium coryli*, and *Komagataella phaffii* (syn. *Pichia pastoris*) are defined as Crabtree negative (respiring) yeasts[8].

It is assumed that the Crabtree phenotype first evolved 125–150 million years ago (mya) in the *Saccharomyces* lineage around the same time as flowering plants which provided a sugar-rich niche for yeasts. The occurrence of the overflow metabolism was the first step in the evolution of aerobic fermentation in yeast[8–11]. There are a number of genetic events that contributed to the emergence of the Crabtree effect in the evolutionary history: whole genome duplication[12,13], horizontal transfer of the *URA1* gene[14], rewiring of the transcriptional network[15,16], loss of respiratory complex I[17], and duplication of hexose transporter genes[18]. Besides this evolutionary development in the *Saccharomyces* lineage, the Crabtree-positive phenotype has evolved several times independently in yeasts[16,19].

Crabtree-positive yeasts have several genes for low-affinity hexose transporters enabling high transport rates at high sugar concentration. On the contrary, Crabtree-negative yeasts usually have only one or two low-affinity *HXT* genes but several high-affinity transporters[20]. Accordingly, lower specific glucose uptake rates in Crabtree-negative yeasts prevent the overflow metabolism that causes reduced biomass yield and ethanol production at high glucose concentrations under aerobic conditions in Crabtree-positive yeasts.

*K. phaffii* is a canonical Crabtree-negative yeast and able to ferment in oxygen-limited conditions[21]. It possesses two high affinity and two low-affinity glucose transporters similar to other respiratory yeasts such as *Kluyveromyces lactis*, *Hansenula polymorpha*, and *Scheffersomyces stipitis*[20]. Consequently, the specific glucose uptake rate at maximum specific growth rate in *K. phaffii* is 8-fold lower[22] in comparison to respiro-fermentative *S. cerevisiae*[23].

Here we show that glycolysis and fermentation are controlled by a single transcription factor (TF) in *K. phaffii*, and that its overexpression is sufficient to convert the Crabtree phenotype, indicating that this regulatory switch in phenotype could have easily occurred during evolution. We characterize the Crabtree phenotype of mutant *K. phaffii* strains and compare it to other Crabtree negative and positive yeasts. Additionally, we investigate transcriptomes and central metabolic fluxes to characterize the physiological responses to overexpression and deletion of the transcription factor. As we show that *K. phaffii* switches to a respiro-fermentative phenotype by altered expression of a single gene, it will be important to investigate in future if *K. phaffii* might be a better model for the Warburg effect than *S. cerevisiae*.

## Results

### A single TF enhances glucose uptake and ethanol production.
Recently, we identified a transcription factor as an activator of the glyceraldehyde 3-phosphate dehydrogenase promoter[24] which has sequence similarity to *S. cerevisiae* Gal4. Putative Gal4 specific transcription factor (TF) binding sites were found in all glycolytic promoters of *K. phaffii*, and overexpression of the *GAL4*-like TF gene led to growth defects in a *K. phaffii* strain when grown in glucose surplus. These results prompted us to dissect the role of this transcription factor on central carbon metabolism of *K. phaffii* as it is obvious that it is not involved in galactose metabolism since *K. phaffii* cannot metabolize galactose.

Overexpression (OE) of this TF changed specific rates of growth, glucose consumption, and ethanol production markedly, as well as biomass and ethanol yields (Table 1 and Supplementary Figure 1), and led to a 1.7-fold higher glucose uptake rate and a 10-fold elevated ethanol yield in shake flask cultivations. Together with the increased ethanol production, the overexpression strain exhibited a diauxic growth profile typical for Crabtree-positive yeasts (Fig. 1). The biomass yield on glucose ($Y_{X/S}$) of the overexpression strain was 1.9-fold lower at the expense of ethanol formation. In contrast, no ethanol was produced by the deletion strain and the specific glucose uptake rate was 2.2-fold lower compared to the control strain, further underlining the role of the Gal4-like transcription factor in the regulation of glucose metabolism. Based on its obvious role in glycolysis and fermentation we decided to allocate a novel name to this transcription factor and suggest *CRA1*, referring to its impact on the Crabtree phenotype of *K. phaffii*.

### A glucose pulse triggers fermentation in the *CRA1* OE strain.
To verify that the *CRA1* overexpression strain is a true Crabtree-positive yeast we investigated its short- and long-term Crabtree phenotype. The short-term Crabtree effect is defined as the response of a non-fermenting culture at steady-state to an excess glucose pulse. As described by Hagman and coworkers[25], control and *CRA1* overexpression strains were grown in steady-state chemostat at a dilution rate $D = 0.1 \, h^{-1}$, followed by a glucose pulse of $C_{Glucose} = 9–11 \, g \, L^{-1}$. We also performed the same experiments with *S. cerevisiae* and *Kluyveromyces marxianus* as internal controls, and verified that our data is comparable to Hagman et al[25]. During the first hour after the glucose pulse biomass of the control strain increased, while it remained unchanged in the cultures of the *CRA1* overexpression strain. Glucose consumption profiles demonstrated the differences in short-term Crabtree phenotype between the control and *CRA1* overexpression strains. While they had a similar specific glucose consumption rate during the steady-state cultivation, it was 1.2-fold higher in the overexpression strain within the first 20 min after the glucose pulse and then decreased (Fig. 2a and Supplementary Table 1). Similarly, higher specific glucose uptake rates were achieved in comparison to other Crabtree negative and positive yeasts except *Vanderwaltozyma polyspora* (Fig. 2a). In addition, the ethanol concentration produced by the *K. phaffii* *CRA1* overexpression strain increased to $0.23 \, g \, L^{-1}$ after the first 20 min which is comparable to other short-term Crabtree-positive yeasts and reached $0.35 \, g \, L^{-1}$ during the first hour after the glucose pulse which is a typical behavior for fermenting yeasts (Fig. 2a and Supplementary Table 1). The specific ethanol

**Table 1 Overexpression or deletion of the *GAL4*-like TF gene changes the phenotype of *K. phaffii***

| Strain[a] | $\mu$[b] [h$^{-1}$] | $q_s$[c] [g g$^{-1}$ h$^{-1}$] | $q_{EtOH}$[d] [g g$^{-1}$ h$^{-1}$] | $Y_{X/s}$[e] [g g$^{-1}$] | $Y_{EtOH/s}$[f] [g g$^{-1}$] |
|---|---|---|---|---|---|
| *K. phaffii* - control | 0.27 ± 0.00 | 0.53 ± 0.05 | 0.00 ± 0.00 | 0.51 ± 0.04 | 0.03 ± 0.02 |
| *K. phaffii GAL4*-like overexpression | 0.22 ± 0.01 | 0.91 ± 0.03 | 0.28 ± 0.05 | 0.24 ± 0.01 | 0.31 ± 0.03 |
| *K. phaffii Δgal4*-like | 0.15 ± 0.00 | 0.24 ± 0.01 | 0.00 ± 0.00 | 0.61 ± 0.02 | 0.00 ± 0.00 |

[a]Mean values of duplicate shake flask experiments are presented. ± indicates the standard deviation.
[b]Specific growth rates
[c]Specific glucose consumption rates
[d]Specific ethanol production rates
[e]Yields of biomass on the substrate
[f]Yields of ethanol on the substrate

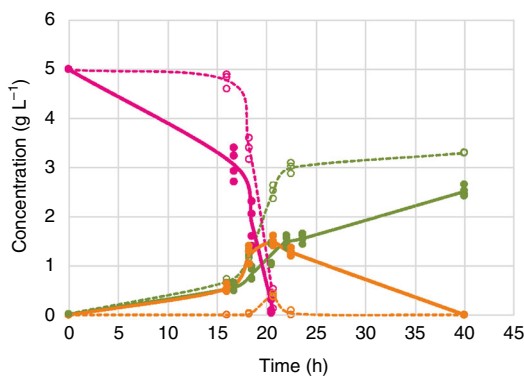

**Fig. 1** *GAL4*-like overexpression strains consume glucose faster and produce ethanol. Biomass dry weight (green), glucose consumption (magenta) and ethanol production (orange) profiles of the control strain (dashed line) and *GAL4*-like overexpression mutants (solid line). Values of 3 different independent transformants of the control (empty circles) and 4 overexpressing the *GAL4*-like TF (filled circles) are shown

production rate of the overexpression strain was in the range of other Crabtree-positive yeasts.

CO$_2$ production of yeast has two primary sources: respiration and fermentation. After the glucose pulse, the *K. phaffii CRA1* overexpression mutant reached higher CO$_2$ evolution rates (CER) compared to the control strain, as 2 moles of CO$_2$ per mole O$_2$ are produced during fermentation different to respiration in which 1 mole O$_2$ consumption corresponds to 1 mole CO$_2$. Following the glucose pulse, CER of *K. phaffii CRA1* overexpression mutants reached 4.53 mmol g$^{-1}$ h$^{-1}$ which was 1.7-fold higher compared to the control strain as well as other Crabtree-negative yeasts (Fig. 2b). These findings fit well to the data from other Crabtree-positive yeasts where CER varied from 3.2–7.0 mmol g$^{-1}$ h$^{-1}$ [25] (Fig. 2b and Supplementary Table 1).

**The long-term Crabtree effect is triggered by *CRA1* OE.** The long-term Crabtree effect is defined as the ability of a yeast strain to ferment glucose to ethanol under aerobic conditions in a glucose-limited chemostat. The critical dilution rate ($D_{crit}$) is defined as the strain-specific threshold value at which the yeast metabolism switches from respiratory state to respiro-fermentative state where ethanol formation starts. In other words, above $D_{crit}$, ethanol is produced as the glucose flux exceeds the critical value towards the TCA cycle. A Crabtree-negative yeast, however, grows in a respiratory state and is washed out without ethanol formation when the maximum dilution rate ($D_{max}$) is exceeded. We conducted chemostat cultivations to determine the critical dilution rate threshold for the *K. phaffii CRA1* overexpression strain (Fig. 3 and Supplementary Table 2).

With the increase in dilution rate ($D$), the specific glucose uptake rates increased accordingly, and similar specific glucose uptake rates were obtained for the control and the *CRA1* overexpression strains until $D = 0.25$ h$^{-1}$. The control strain could grow in fully respiratory state without any ethanol formation until the dilution rate reached $D_{max} = 0.4$ h$^{-1}$ and was washed out when the dilution rate was further increased to 0.45 h$^{-1}$. However, the *CRA1* overexpression mutant displayed a clear long-term Crabtree phenotype above $D = 0.25$ h$^{-1}$: the cell concentration decreased, and ethanol production began at $D_{crit} = 0.27$ h$^{-1}$. When compared to the control strain, a 1.7-fold higher specific glucose consumption rate was achieved by the overexpression strain at the critical dilution rate ($D_{crit} = 0.27$ h$^{-1}$) indicating a higher glycolytic flux. Accordingly, ethanol production started, showing that a fraction of metabolic activity shifted from respiratory metabolism to respiro-fermentative metabolism. When the dilution rate was further increased to $D = 0.32$ h$^{-1}$, the cells were washed out and could not reach a new steady-state showing that $D_{crit} = D_{max} = 0.27$ h$^{-1}$ for the *CRA1* overexpression strain.

Comparison of several yeast species based on their specific glucose consumption rates, respiratory quotients, specific ethanol production rates, CER, oxygen uptake rates (OUR), and maximum specific growth rates revealed that there is a universal cut-off value for glucose consumption rates which is around 10–15 C-mmol g$^{-1}$ h$^{-1}$ [8]. This is the cut-off value where Crabtree-positive yeasts exhibit an imbalanced catabolic and anabolic metabolism, due to a loss of coordination between glucose consumption and growth. This results in an overflow in carbon metabolism, leading to ethanol production. Indeed, the *K. phaffii CRA1* overexpression strain started to produce ethanol when $q_S$ exceeded 15.8 C-mmol g$^{-1}$ h$^{-1}$ (Fig. 3 and Supplementary Table 2) as opposed to the control strain where the balance could be preserved between glucose consumption rate and carbon flux towards catabolic and anabolic pathways at even higher growth rates.

**The *CRA1* OE strain classifies as Crabtree-positive yeast.** Crabtree phenotype defines whether a yeast can perform simultaneous respiration and fermentation under aerobic conditions. However, not every Crabtree-positive yeast shows this phenotype in the same range. Hagman and coworkers[26] classified 11 different yeast genera (including 55 different strains of 34 species) into 4 groups, based on their ethanol and biomass yields on glucose, specific glucose consumption rates and specific growth rates in a batch culture. Among these, group 1 includes the Crabtree-negative yeasts and groups 2–4 include Crabtree-positive yeasts. We used this classification as an indicator to rate the Crabtree phenotype of the *CRA1* overexpression strain among other yeasts and divided yeasts into 2 different groups: group 1 includes the Crabtree-negative yeasts (including 15 different strains of 8 species) and group 2 (including 40 different

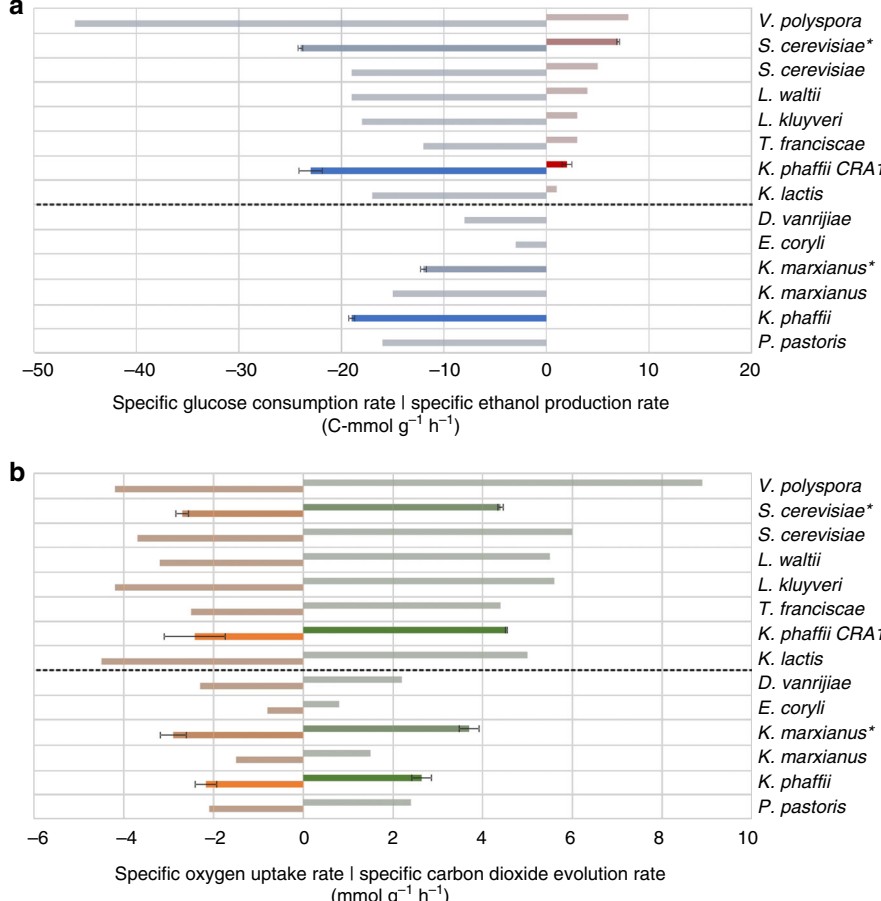

**Fig. 2** The *CRA1* overexpression strain exhibits a short-term Crabtree phenotype. *K. phaffii* strains were grown in chemostat at $D = 0.1\,h^{-1}$ in steady state, followed by a glucose pulse of 9-11 g L$^{-1}$. Data of *K. phaffii* control, overexpression strain, *Saccharomyces cerevisiae* CEN.PK113-5D and *Kluyveromyces marxianus* CBS712 (indicated with an asterisk) are shaded dark. The data of other yeast species were retrieved from Hagman et al.[25] and shaded light. They comprise the Crabtree positive species (above the dashed line) *Vanderwaltozyma polyspora*, *S. cerevisiae*, *Lachancea waltii*, *Lachancea kluyveri*, *Torulaspora franciscae*, *Kluyveromyces lactis*, and the Crabtree negative species (below the dashed line) *Debaryomyces vanrijiae*, *Eremothecium coryli*, *K. marxianus* and *P. pastoris* (syn. *Komagataella* sp.). **a** Comparison of specific glucose consumption (blue) and ethanol formation (red) rates. The specific glucose uptake rate of the *CRA1* overexpression strain was higher compared to the control strain and other Crabtree-negative yeasts. Ethanol production by the *CRA1* overexpression strain was comparable to other short-term Crabtree-positive yeasts, while the control strain did not produce ethanol, same as the other Crabtree-negative yeasts. **b** O$_2$ uptake (orange) and CO$_2$ evolution rates (green) among other yeasts based on their short-term Crabtree effect. Calculated CO$_2$ evolution rate values for the *CRA1* overexpression strain were in the range of other short-term Crabtree-positive yeasts (see ref. [25]). Mean values of the duplicate bioreactor cultivations for the control and overexpression *K. phaffii* strains are presented. Error bars represent the standard deviation (±)

strains of 26 species from groups 2–4[26]) contains the Crabtree-positive yeasts (having a Crabtree phenotype in different ranges) (Fig. 4). The *K. phaffii* control strain fell into Group 1 with other Crabtree-negative yeasts. The *CRA1* overexpression strain, however, was classified as a Crabtree-positive yeast in Group 2.

**Upregulation of glycolytic genes causes metabolic imbalance.** For an in-depth analysis of the effect of the Cra1 TF on regulation of the glycolytic pathway, we conducted RNASeq and $^{13}$C labeled metabolic flux analysis (MFA) (Fig. 5). Transcript levels of glycolytic (*HXK1, PGI1, PFK1, PFK2, FBA1-1, TDH3, ENO1, GPM1, CDC19*) and fermentative genes (*PDC1, ADH2*) were upregulated in the *CRA1* overexpression strain compared to the control strain while the genes responsible for the TCA cycle (*CIT1, ACO1, IDP1, KGD2, FUM1, MDH1, MDH3*) remained unchanged. In agreement with the RNASeq data, the glycolytic flux was more than twofold higher in the *CRA1* overexpression strain. The split ratio (%) at the glucose-6-phosphate node was 85/15 (± 1.5)

(glycolysis/pentose phosphate pathway) in the *CRA1* overexpression strain, compared to 44/56 (± 0.1) in the control strain. Around the pyruvate node, the ratio (%) between fermentation and respiration was 68/32 (± 13) (ethanol/acetyl CoA) in the overexpression strain compared to 17/83 (± 21) in the control. Comparing specific carbon fluxes, a decrease of pentose phosphate pathway and TCA cycle fluxes contributed to the 10-fold increase of specific fermentation flux. This finding is in agreement with the situation in *S. cerevisiae*, where higher glycolytic fluxes cause a decrease in the TCA fluxes[8,27,28]. The increase of specific NADH production in lower glycolysis between control and overexpression strain of 4.7 mmol g$^{-1}$ h$^{-1}$ balances well with the increased specific NADH consumption in ethanol formation of 5.4 mmol g$^{-1}$ h$^{-1}$, and thus restores the redox balance in the *CRA1* overexpression strain through respiratory-fermentative metabolism. These data further support the hypothesis that a single gene controls the development of a Crabtree positive strain by upregulating all relevant glycolytic and fermentative genes.

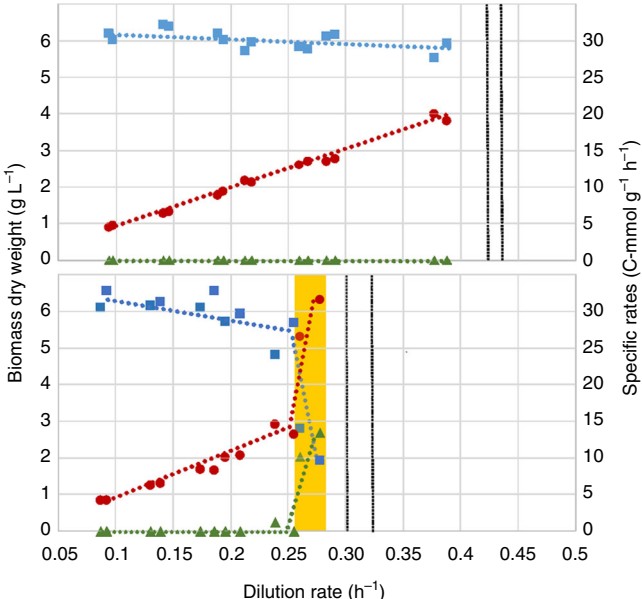

**Fig. 3** The long-term Crabtree effect is triggered by overexpression of *CRA1*. The control strain exhibited a fully respiratory metabolism and was washed out after $D_{max} = 0.4\,h^{-1}$ (upper panel). The metabolic activity of the *CRA1* overexpression strain, however, shifted from respiratory metabolism to respiro-fermentative metabolism and started to produce ethanol when $D_{crit} = 0.27\,h^{-1}$ was achieved (lower panel). Above $D_{crit}$ glucose consumption rates increased sharply while biomass concentrations dropped. Mean values of biomass dry weight concentration (squares), specific glucose consumption (circles) and specific ethanol production rates (triangles) of the control and *CRA1* overexpression strain at different dilution rates during the chemostats are given (chemostats were performed in duplicates). Gray and black lines show the dilution rates of the biological replicates at which the cells were washed out

In addition to the glycolytic and fermentative pathway genes mentioned above, several other genes were also found to be upregulated in the *CRA1* overexpression strain. Of these, the genes responsible for thiamine biosynthesis (*THI20*, *THI4*, *THI11*) were upregulated in the *CRA1* overexpression strain (and downregulated in the Δ*cra1* strain) compared to the control. *THI7* was downregulated in the Δ*cra1* strain in comparison to the control in addition to the downregulated thiamine genes mentioned above. As thiamine is required for several enzymes of the central carbon metabolism like pyruvate decarboxylase, pyruvate dehydrogenase or transketolase, increased transcript levels of thiamine biosynthesis genes were expected. In *S. cerevisiae*, disruption of *THI2* and *THI3* was shown to impair pyruvate decarboxylase activity, enhancing pyruvate production and decreasing ethanol formation[29]. In this yeast, *PDC* and thiamine biosynthesis genes are linked by a common transcriptional activator, Pdc2[30]. Up to now, no homolog of *PDC2* was identified in *K. phaffii*, but it is interesting that a convergent co-regulation by a Gal4-like TF is present.

## Discussion

When we analyzed the promoter sequences of the glycolytic genes in *K. phaffii*, putative binding sites of a Gal4-like TF were found[24]. *K. phaffii* does not possess a galactose metabolism[31], therefore it is obvious that this TF is not involved in galactose utilization. As the previous gene designation (*GAL4*) was solely based on its sequence similarity to the ortholog in *S. cerevisiae*, we attributed a new short name to the gene PP7435_Chr1-1522 as *CRA1* to avoid any confusion related to the different roles of this

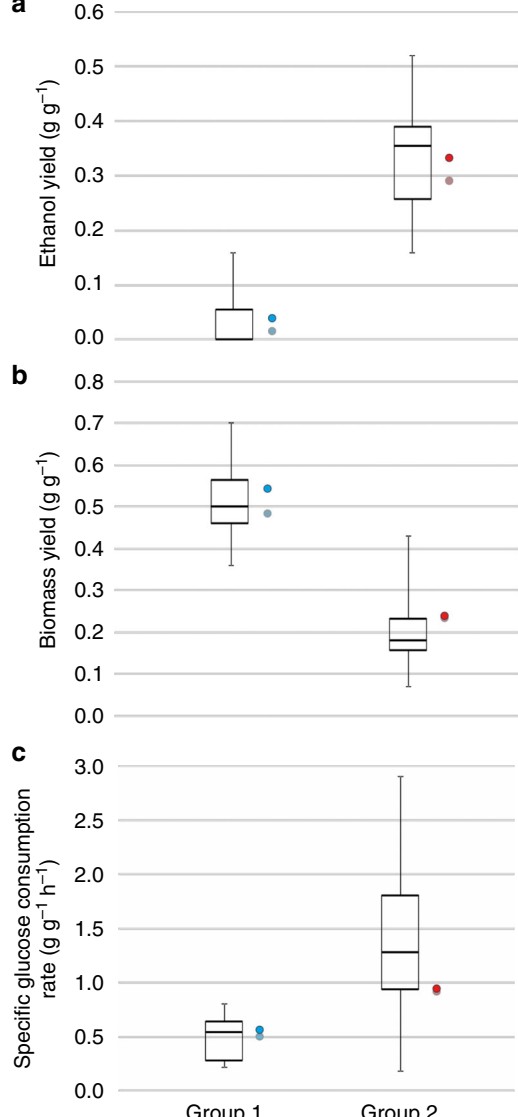

**Fig. 4** The *CRA1* overexpression strain classifies as a Crabtree-positive yeast. Classification of yeasts based on **a** ethanol yield, **b** biomass yield, and **c** specific glucose consumption rate reveals that the control strain, *K. phaffii* (blue), belongs to Group 1 with other Crabtree-negative yeasts (*Eremothecium* and *Kluyveromyces* genera, see ref. [26]). *K. phaffii CRA1* overexpression strain (red), however, is classified as a Crabtree-positive yeast with other positive yeast species (*Lachancea*, *Torulaspora*, *Zygotorulaspora*, *Vandervaltozyma*, *Tetrapisispora*, *Saccharomyces*, *Kazachstania*, *Naumovozyma*, and *Nakaseomyces* genera). Values of the duplicate experiments for the control and overexpression *K. phaffii* strains are presented. The box plots are based on the data of ref. [25]. The box spans the interquartile range (second and third quartile), middle line represents the median value and whiskers show the minimum and maximum values

gene in *K. phaffii* and *S. cerevisiae*, respectively. *K. phaffii* is a canonical Crabtree-negative yeast and its glucose uptake capacity is limited[13]. However, the overexpression of *CRA1* resulted in a significant increase in glucose uptake rate and ethanol yield compared to the control strain (Fig. 1 and Table 1). Given the fact that the specific glucose uptake rate of a Δ*cra1* strain was 2.2-fold lower compared to the control strain, it appears that *CRA1* has a regulatory role in glucose metabolism of *K. phaffii*. Indeed, the *CRA1* overexpression strain was classified as a Crabtree-positive yeast contrary to the control strain (Fig. 4). Both short- and long-

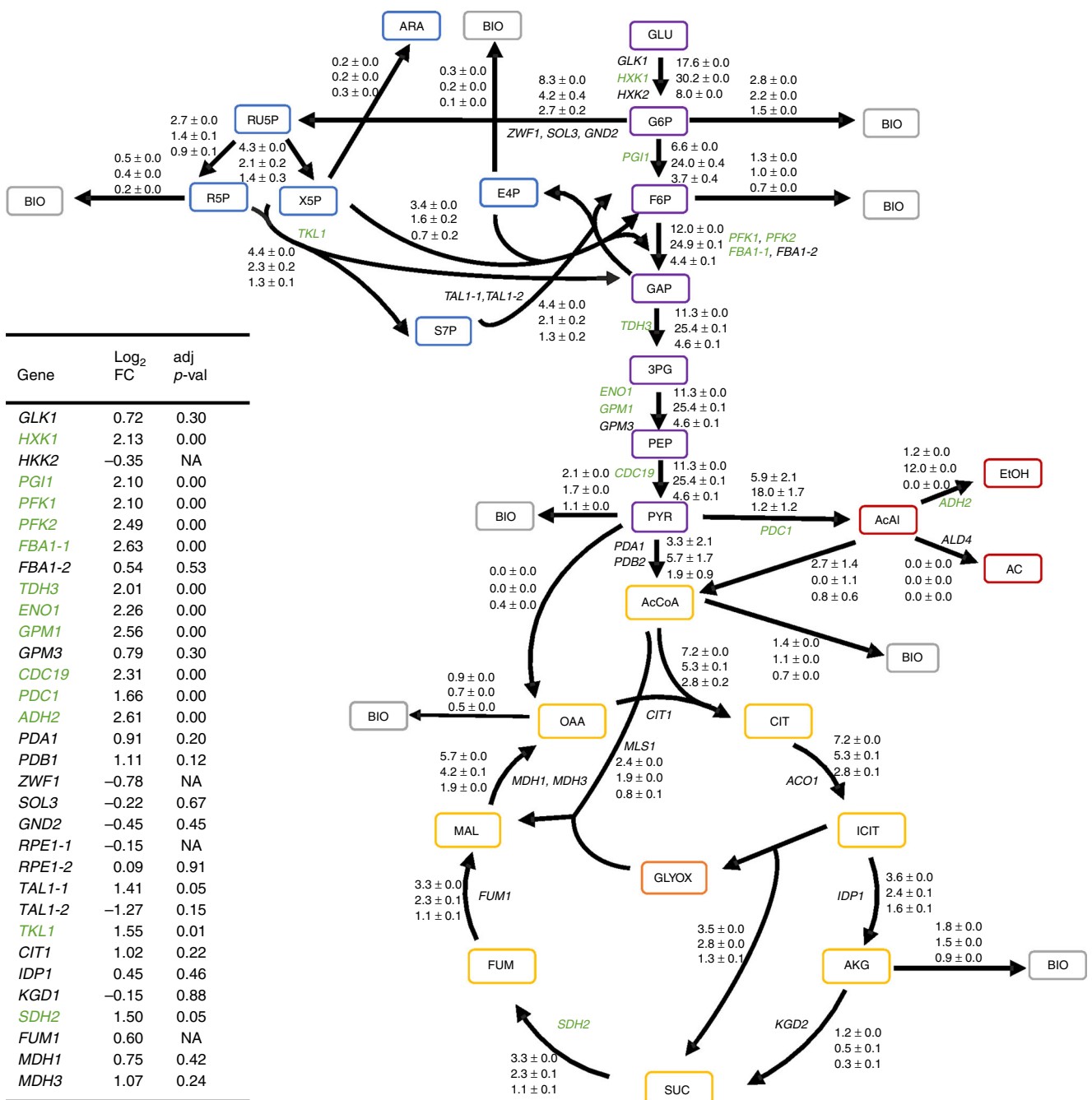

| Gene | Log₂ FC | adj p-val |
|------|---------|-----------|
| GLK1 | 0.72 | 0.30 |
| HXK1 | 2.13 | 0.00 |
| HKK2 | −0.35 | NA |
| PGI1 | 2.10 | 0.00 |
| PFK1 | 2.10 | 0.00 |
| PFK2 | 2.49 | 0.00 |
| FBA1-1 | 2.63 | 0.00 |
| FBA1-2 | 0.54 | 0.53 |
| TDH3 | 2.01 | 0.00 |
| ENO1 | 2.26 | 0.00 |
| GPM1 | 2.56 | 0.00 |
| GPM3 | 0.79 | 0.30 |
| CDC19 | 2.31 | 0.00 |
| PDC1 | 1.66 | 0.00 |
| ADH2 | 2.61 | 0.00 |
| PDA1 | 0.91 | 0.20 |
| PDB1 | 1.11 | 0.12 |
| ZWF1 | −0.78 | NA |
| SOL3 | −0.22 | 0.67 |
| GND2 | −0.45 | 0.45 |
| RPE1-1 | −0.15 | NA |
| RPE1-2 | 0.09 | 0.91 |
| TAL1-1 | 1.41 | 0.05 |
| TAL1-2 | −1.27 | 0.15 |
| TKL1 | 1.55 | 0.01 |
| CIT1 | 1.02 | 0.22 |
| IDP1 | 0.45 | 0.46 |
| KGD1 | −0.15 | 0.88 |
| SDH2 | 1.50 | 0.05 |
| FUM1 | 0.60 | NA |
| MDH1 | 0.75 | 0.42 |
| MDH3 | 1.07 | 0.24 |

**Fig. 5** $^{13}$C metabolic flux analysis and RNASeq explain the overflow metabolism. Glycolysis and fermentation correlate with pathway gene upregulation in the *CRA1* overexpression strain. Absolute specific flux rates per carbon atom (C-mmol g$^{-1}$h$^{-1}$) of control, *CRA1* overexpression and $\Delta cra1$ strains are given from top to bottom. For reversible fluxes, only net fluxes are shown. Gene names of the corresponding reactions are colored green for transcriptionally upregulated, and black for unregulated genes in the *CRA1* overexpression strain compared to the control. The insert table displays the respective log$_2$ fold changes and adjusted *p*-values. NA means data did not meet filter criteria for *p*-value calculation (see Methods). Mean values of triplicate experiments for the control, overexpression and knock-out *K. phaffii* strains are presented. ± represents the standard deviation. Values of each reaction refer to the total number of carbon atoms involved in this reaction. For clarity of the figure, fluxes to CO$_2$ are not shown

term Crabtree phenotypes were triggered in the *CRA1* overexpression strain (Figs 2–3) indicating an overflow of the glycolytic flux due to the overexpression of *CRA1*. Upregulation of the glycolytic genes and an increase in glucose uptake rate resulted in higher glycolytic flux towards lower glycolysis, obviously causing an imbalance between respiration and biomass formation (Fig. 5). This higher and imbalanced glycolytic flux then caused an overflow metabolism which led to an increased flux through the fermentative pathway, *i.e.* ethanol production which is a typical

phenotype for the majority of the respiro-fermenting yeasts including species from *Saccharomyces*, *Kluyveromyces*, *Lachancea*, *Torulaspora*, *Zygotorulaspora*[8].

Several evolutionary mechanisms have been proposed to understand how the Crabtree phenotype emerged. The Crabtree effect arose several times in yeast evolution[16,19]. In the *Saccharomyces* lineage, the extent of the Crabtree effect in each species relates to its position in the phylogenetic tree[9]. A whole genome duplication (WGD) event approximately 100 mya ago[13] and a

global rewiring of the rapid growth elements[11,13] that changed the regulation of genes associated with respiration and rapid growth, are the major evolutionary steps that contributed to the ability of ethanol production. It has been suggested that the Crabtree effect originated approximately 125 mya with the horizontal transfer of the *URA1* gene which enabled yeasts to grow anaerobically after the loss of respiratory chain complex I[14,17]. However, our data suggest that deregulation of a single transcription factor can lead to the invention of the Crabtree effect, without the requirement of anaerobic growth or loss of respiratory chain complex I. This was also suggested as a possible scenario by Dashko and coworkers[9] based on the data of Hagman et al.[26] They suggest that the initial evolutionary driving force in sugar-rich environments may have been connected to regular exposure to oxygen-limited niches, so that yeasts developed strengthened glycolytic and fermentation pathways for anaerobic growth. This may have been the later basis for the development of aerobic fermentation. Previously we showed that in hypoxic conditions the fermentative pathway is switched on in *K. phaffii*, and glycolysis is upregulated[21]. This means, *K. phaffii* ferments with higher glycolytic flux and higher glucose uptake rates in oxygen limitation, but not in aerobic conditions. By overexpression of *CRA1*, the joint regulation of glycolysis and fermentation leads to the Crabtree positive phenotype, without the need of any other mutation such as the change of *URA1* or the loss of respiratory complex I. This suggests that aerobic fermentation may have occurred very early in yeast evolution, even before anaerobic growth was fully established.

Dashko and coworkers[9] also pointed out the role of hexose transporters in glucose utilization and argue that multiplication of hexose transporters could be one of the molecular backgrounds for the increased glucose consumption ability. Interestingly, the hexose transporter genes were not upregulated by *CRA1* overexpression indicating that the increase in the glucose uptake rate is driven by the enhanced glycolytic flux rather than the upregulation of hexose transporter genes. We calculated the kinetic parameters, $K_m$ and $V_{max}$, and a two-tailed Student's *t*-test showed that there is no statistically significant difference (*p*-value = 0.68) between the $K_m$ values of the control ($K_m$ = 14.1 ± 3.8 mM) and the *CRA1* overexpression strain ($K_m$ = 12.8 ± 3.3 mM). $V_{max}$ values, however, differed significantly (control: 0.85 g g$^{-1}$ h$^{-1}$, *CRA1* overexpression: 1.49 g g$^{-1}$ h$^{-1}$ h, *p*-value = 0.002).

According to Martchenko et al.[32] Gal4 was recruited (together with Mig1) late in ascomycete yeast evolution to regulate the Leloir pathway for galactose utilization. In yeasts that originate from ancestors prior to this transition, such as *C. albicans*, *Debaryomyces hansenii*, or *Schizosaccharomyces pombe*, the Leloir pathway genes are controlled by Cph1. *K. lactis* and *Naumovozyma castellii* (syn. *Saccharomyces castellii*) appear to mark the transition, having both Gal4 and Mig1 as well as Cph1 binding sites in the Gal gene promoters, while later in evolution, *Saccharomyces* species have lost Cph1 binding sites. Deletion of the *GAL4*-like gene in *C. albicans* indicated that it is involved in regulation of glycolysis[32,33]. *GAL4* of *S. cerevisiae* is regulated at the transcriptional level and by the accessory proteins Gal80 and Gal3. *K. phaffii* and other Crabtree-negative yeasts do not possess homologs of Gal80 and Gal3[34]. In *K. phaffii* the *CRA1* gene is transcriptionally upregulated in glucose surplus[35] and under severe glucose limitation, when compared to limited but reasonably good supply with glucose like in the chemostat[36,37].

We analyzed glycolytic and *GAL* gene promoters of *S. cerevisiae*, *Kazachstania naganishii*, *N. castellii*, *Zygosaccharomyces rouxii*, *L. kluyveri*, *Eremothecium gossypii* (syn. *Ashbya gossypii*), *K. lactis*, *C. albicans*, *Sugiyamaella lignohabitans*, *D. hansenii*, *Y. lipolytica*, and *Ogataea parapolymorpha* for the presence of Gal4 binding sites (Supplementary Table 3). A high number of Gal4

binding sites (around 20–30 in total) in promoters of glycolytic genes were found in *K. phaffii*, *E. gossypii*, *D. hansenii*, *K. naganishii*, *S. lignohabitans*, *O. parapolymorpha*, and *C. albicans*, while the other yeast genomes contain lower numbers of binding sites in glycolytic gene promoters. *K. naganashii* and *Z. rouxii* apparently mark transitions with similar, rather high numbers of binding sites in both pathways. With these two exceptions, a high number of Gal4 binding sites in galactose utilization gene promoters correlates with a decreased number in glycolytic promoters, supporting the general hypothesis that during evolution Gal4 switched its function from a generalist TF regulating the central carbon metabolism to a specialist TF responsible for the galactose metabolism[34]. We overexpressed the native *CRA1* homolog (annotated as *GAL4*) in *S. lignohabitans* to test whether it acts as a generalist TF on the central carbon metabolism in this yeast as well. Indeed, overexpression of *GAL4* resulted in a 1.3-fold increase in specific glucose uptake rate and a 1.5-fold increase in specific glycerol production rate, respectively (Supplementary Table 4). Glycerol is a by-product of the central carbon metabolism, produced to sustain the redox balance similar to ethanol. These data support the hypothesis that the Gal4 ortholog acts as a generalist TF of glucose metabolism in Crabtree-negative yeasts.

Up to now, the evolution of the Crabtree phenotype has been linked to several complex genetic and metabolic changes i.e. whole genome duplication, loss of respiratory complex I, and rewiring of the transcriptional network. Here we show that a single transcription factor controls glycolysis and fermentation of the Crabtree-negative yeast *K. phaffii*. Its overexpression is sufficient to convert *K. phaffii* to a Crabtree-positive yeast, indicating that a single mutation may have been sufficient as a first evolutionary event towards respiro-fermentative metabolism. Apart from shedding new light on the evolution of the Crabtree effect in yeasts, this novel *K. phaffii* strain may serve as a better model to study the Warburg effect of cancer cells.

## Methods

**Strains.** All strains were derived from the parental strain *Komagataella phaffii* CBS7435. *CRA1* overexpression strains were constructed by using Golden*PiCS*[38]. The Δ*cra1* strain was obtained by the split marker cassette method[39]. In part of the strains the eGFP gene (under control of the glyceraldehyde-3-phosphate dehydrogenase promoter (P$_{GAP}$)) was used as a reporter of *CRA1* based induction of glycolytic genes. These strains were used in RNASeq and $^{13}$C metabolic flux analysis. In all the transformations, at least 3 random transformants were selected as biological replicates for further shake flask experiments to minimize the risk of accidental spurious mutations that could cause phenotypic changes. We also used two different background strains to overexpress *CRA1* (Table 2) which resulted in the same phenotypic changes. One representative transformant was used for RNASeq and $^{13}$C metabolic flux analysis. Two different biological replicates for each strain were used in chemostat cultures.

A BLAST analysis was performed to identify the putative *GAL4* ortholog of *S. lignohabitans* (NCBI RefSeq Accession Number: XP_018737608.1 [https://www.ncbi.nlm.nih.gov/protein/XP_018737608.1/]). After PCR amplification from *S. lignohabitans* strain CBS10342 the gene was cloned into an integration vector under control of P$_{GAP}$ of *S. lignohabitans* and transformed into the same strain[40].

**Shake flask cultivations.** All shake flask experiments were performed in 200 mL wide neck shake flasks with 20 mL M2 medium containing (g L$^{-1}$) 5 glucose, 3.15 (NH$_4$)$_2$HPO$_4$; 0.49 MgSO$_4$.7H$_2$O; 0.8 KCl; 0.0268 CaCl$_2$.2H$_2$O; 22 citric acid monohydrate; 1.47 mL PTM; 2 mL biotin (0.2 g L$^{-1}$). Cells grown in YP medium for 24–26 h were washed and transferred into M2 medium. Samples were collected during the exponential growth phase for the characterization of the strains (in duplicates), calculation of the kinetic parameters ($V_{max}$ and $K_m$) of the glucose transporters (in triplicates), $^{13}$C metabolic flux analysis (in triplicates) and RNASeq analysis (in triplicates).

**Chemostat cultivations for long-term and short-term Crabtree effect.** All chemostat cultivations were performed in duplicate in DASGIP (Jülich, Germany) bioreactors with a working volume of 1 L. pH was controlled at 5.00 with 12.5% ammonia solution. Dissolved oxygen concentration was kept above 30% with a constant airflow of 16 L h$^{-1}$. One mL cryostock of *K. phaffii* strains were inoculated

**Table 2 *K. phaffii* strains used in the study**

| Referred in the manuscript as | Strain | Used in | Reference |
|---|---|---|---|
| Control strain | *K. phaffii* P$_9$::egfp | RNASeq, Flux analysis | Ata et al.[24] |
| CRA1 overexpression | *K. phaffii* P$_9$::egfp-CRA1-OE | RNASeq, Flux analysis | Ata et al.[24] |
| CRA1 knock-out | *K. phaffii* P$_9$::egfp Δcra1 | RNASeq, Flux analysis | Ata et al.[24] |
| CRA1 overexpression | *K. phaffii* CRA1 overexpression | Chemostats for short- and long-term Crabtree phenotype | This study |
| Control strain | *K. phaffii* – vector control | Chemostats for short- and long-term Crabtree phenotype | This study |

into 100 mL of YPG medium (g L$^{-1}$: 10 yeast extract, 20 soy-peptone, 20 glycerol) including 25 µg L$^{-1}$ Zeocin as preculture and incubated at 25 °C and 180 rpm for approximately 24 h. This culture was inoculated to the bioreactor with an OD$_{600}$ of 0.3 to initiate batch cultivation. Batch medium included (g L$^{-1}$) 2 citrate mono-hydrate, 12.6 (NH$_4$)$_2$HPO$_4$, 0.5 MgSO$_4$.7H$_2$O, 0.9 KCl, 0.022 CaCl$_2$.2H$_2$O, 4.6 mL PTM, 2 ml biotin (0.2 g L$^{-1}$). 10 g L$^{-1}$ and 7.2 g L$^{-1}$ glycerol was added to the batch medium for the investigation of long-term and short-term Crabtree effects, respectively. PTM stock solution contained (g L$^{-1}$) 6.0 CuSO$_4$·5H$_2$O, 0.08 NaI, 3.36 MnSO$_4$·H$_2$O, 0.2 Na$_2$MoO$_4$·2H$_2$O, 0.02 H$_3$BO$_3$, 0.82 CoCl$_2$, 20.0 ZnCl$_2$, 65.0 FeSO$_4$·7H$_2$O and 5.0 mL H$_2$SO$_4$ (95%–98%). Following the batch end, chemostat cultivation was started.

Investigation of the short-term Crabtree effect of the *K. phaffii* control and overexpression strains was performed in 700 mL medium at 25 °C with a similar experimental design made by Hagman and coworkers[25]. Cells were grown in chemostat cultivation at $D = 0.1$ h$^{-1}$ at fully respiring conditions. After five residence times until steady state was achieved, a glucose pulse was applied to the cultures to a final concentration of 9–11 g L$^{-1}$ ($t = 0$ min) and samples were taken every 20 min. Chemostat medium contained (g L$^{-1}$) 0.36 citrate monohydrate, 7.92 glucose monohydrate, 3.13 (NH$_4$)$_2$HPO$_4$, 0.14 MgSO$_4$.7H$_2$O, 0.36 KCl, 0.005 CaCl$_2$.2H$_2$O, 0.35 ml PTM, 0.29 biotin (0.2 g L$^{-1}$). Batch and chemostat cultivations for *S. cerevisiae* CEN.PK113-5D and *K. marxianus* CBS712 were performed in the same chemostat medium but supplemented with vitamin solutions: (mg L$^{-1}$) 0.04 biotin, 0.72 calcium pantothenate, 0.72 nicotinic acid, 18.0 myo-inositol, 0.72 thiamine HCl, 0.72 pyridoxine, 0.14 para-aminobenzoic acid. Additionally, uracil with a concentration of 54 mg L$^{-1}$ was used for cultivation of uracil auxotrophic *S. cerevisiae* CEN.PK113-5D.

Evaluation of the long-term Crabtree effect of the strains was carried out in 350 mL medium at 30 °C in fully respiring conditions. Chemostat cultivation was started with a dilution rate ($D$) 0.1 h$^{-1}$ and increased incrementally until ethanol formation set in and increased further until cells were washed out. Samples were collected for each dilution rate after five residence times when steady state was reached. Chemostat medium contained (g L$^{-1}$) 0.5 citrate monohydrate, 11.0 glucose monohydrate, 4.35 (NH$_4$)$_2$HPO$_4$, 0.2 MgSO$_4$.7H$_2$O, 0.5 KCl, 0.007 CaCl$_2$.2H$_2$O, 0.48 mL PTM, 0.4 mL biotin (0.2 g L$^{-1}$).

**Analysis of bioreactor samples.** For determination of dry cell weight (DCW), culture broth was centrifuged; pellets were washed with distilled water and dried for 24 h at 100 °C in preweighed glass tubes. Harvested supernatants were used for measurement of extracellular metabolites, and their concentrations were quantified by HPLC analysis[41]. Specific growth rates, glucose uptake rates and metabolite secretion rates were calculated based on the HPLC measurements. CO$_2$ evolution rates (CER) and O$_2$ utilization rates (OUR) were calculated by using an off-gas analyzer.

**$^{13}$C metabolic flux analysis.** Culture broth (corresponding approximately to 15–20 mg total dry biomass) was mixed with 60% (v/v) cold methanol ($< -20$ °C) for quenching and filtered immediately[42]. Metabolites were extracted using boiling ethanol extraction[43].

$^{13}$C labeling patterns of intracellular metabolites were analyzed by GC-QTOFMS[44]. Specific growth, glucose uptake and metabolite secretion rates were calculated during the exponential growth phase. Biomass composition of the control, overexpression and knock-out strains were assumed to be similar to those determined previously by Carnicer and coworkers[45].

The stoichiometric model (Supplementary Table 5) used for the calculation of intracellular fluxes was based on a previously published model of *K. phaffii* central carbon metabolism[46]. OpenFLUX with default settings was used for flux calculation applying the gradient based search algorithm for sensitivity analysis by a Monte Carlo algorithm[47]. Mass distribution values and sensitivity analysis results are provided in Supplementary Table 6. All calculations were done with MATLAB (R2013a, The MathWorks, Inc., Natick, Massachusetts, USA).

**RNASeq analysis.** Pellets from the samples collected during the exponential growth phase were resuspended in 1 mL TRI Reagent (*Sigma*, USA). Cells were disrupted by glass beads using a ribolyser (*MP Biomedicals*, USA) for 40 s at 5.5 ms$^{-1}$. RNA was extracted by chloroform and precipitated with isopropanol. After washing twice with 75% ethanol, RNA was dissolved in nuclease-free water. After DNase treatment (*Invitrogen*, USA), concentration of RNA was measured by Nanodrop and quality of the samples was verified with the Agilent Bioanalyzer 2100 using the RNA 6000 Nano Assay kit (Agilent Technologies, California, USA).

RNA sequencing was performed with poly(A)-enriched mRNAs at Vienna Bio-center Core Facilities GmbH (AT). cDNA libraries were constructed from these poly(A)-enriched mRNAs and sequenced by Illumina HiSeq v4.

**Processing of the RNASeq data.** The resulting unsorted and unaligned BAM files where sorted with samtools sort v1.5, followed by conversion and splitting into paired-end FASTQ files by using picard tools v2.17.3-3-g3bf7fdc-SNAPSHOT SamToFastq (https://broadinstitute.github.io/picard/). Subsequently the files where trimmed and quality checked using TrimGalore v0.4.2 (http://www.bioinformatics.babraham.ac.uk/projects/trim_galore/) with default parameters. As preparation for the count quanti-fication a transcript index was created using the latest *K. phaffii* CBS7435 annotation (FR839628.1, FR839629.1, FR839630.1, FR839631.1, FR839632.1; October 2016) and the kallisto v0.43.0[48] subprogram index. Annotations of the genes were based on the recent curation by Valli et al.[49] which included 5325 open reading frames and 16 mitochondrial genes. Afterwards the read count for each sample was determined using kallisto subprogram quant[50]. For differential expression analysis[51] with R (http://www.R-project.org/) the packages tximport and tximportData[52], readr (https://CRAN.R-project.org/package=readr) and DESeq2 (https://rdrr.io/bioc/DESeq2)[53–55] were used for each comparison. Based on the determination of the differential expression by using DESeq2 some fold-change or *p*-values are not displayed (represented as NA). This result can occur in three cases: Firstly, if all samples of a row have zero counts, secondly, if a sample denotes an extreme count outlier (detected by Cook's distance) and finally a sample row is excluded by automated filtering for having a low mean normalized count[54]. Visualization of the expression analysis results was also carried out with R. Cutoff criteria were a minimum fold-change of 2 and an adjusted *p*-value lower than 0.05 (Supplementary Data 1).

**Characterization of glucose transport kinetics.** $V_{max}$ and $K_m$ values of the glucose transporters of the control and *CRA1* overexpression strains were determined based on Michaelis-Menten kinetics. Specific glucose consumption rates were calculated by using M2 medium with different initial glucose concentrations, i.e. 2.5, 5.0, 10.0, and 20 g L$^{-1}$. Samples were collected during the exponential growth phase. Residual glucose and biomass concentrations were measured to calculate the specific glucose uptake rates. A Lineweaver-Burk plot was used to dissect the Michaelis-Menten equation and determine the kinetic parameters.

**Promoter sequence analysis.** Putative Gal4 transcription factor binding sites in the promoter sequences of the glycolytic and galactose metabolism genes of 13 yeast species were analyzed by MatInspector release 8.4.1 within the Genomatix Suite[56] by using the Matrix Family Library Version 11.0 for fungi. Default settings were left unchanged and matching scores higher than 0.75 were considered. Only positions with a predicted binding site on both DNA strands were considered as true matches. 1000 bp upstream of the genes were considered as the promoter sequences and sequences were retrieved from Genome Resources for Yeast Chromosomes (GRYC), National Center for Biotechnology Information (NCBI) and www.pichiagenome.org.

**Reporting Summary.** Further information on research design is available in the Nature Research Reporting Summary linked to this article.

## Data availability

Raw data of the RNASeq analysis was deposited to the NCBI Gene Expression Omnibus (GEO) repository under accession number [GSE110624]. All other data are available upon reasonable request from the corresponding author (DM).

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

## Acknowledgements

We thank Hannes Rußmayer for his support in sample preparation for GC-QTOFMS analysis of $^{13}$C labeled metabolites. A special thank is due to the late Prof. Jure Piškur for sharing his knowledge and enthusiasm on yeast metabolism and evolution of fermentation. ÖA was awarded scholarships by the Scientific and Technical Research Council of Turkey (TÜBİTAK-BIDEB) and the Austrian Agency for International Cooperation in Education and Research (OeAD), financed by the Austrian Federal Ministry of Science, Research and Economy (BMWFW) (grant no. ICM-2017-06356). This work has been partly supported by the Austrian Federal Ministry for Digital and Economic Affairs (BMDW), the Federal Ministry of Traffic, Innovation and Technology (bmvit), the

Styrian Business Promotion Agency SFG, the Standortagentur Tirol, the Government of Lower Austria and ZIT – Technology Agency of the City of Vienna through the COMET-Funding Program managed by the Austrian Research Promotion Agency FFG, and by the Christian Doppler Research Association, the Austrian Federal Ministry for Digital and Economic Affairs (BMDW) and the National Foundation for Research, Technology and Development (CR). Vienna Business Agency and EQ BOKU VIBT GmbH are acknowledged for providing mass spectrometry and bioreactor instrumentation.

## Author contributions

ÖA contributed to the design of the study and performed the shake flask and bioreactor cultivations, promoter motif analysis, samplings for RNASeq and $^{13}$C labeled metabolic flux experiments, data analysis and interpretation, and drafted the manuscript. CR performed the bioreactor cultivations. NET performed the RNASeq data processing and investigated the statistical significance of the results. MV helped with the *K. phaffii* genome annotation and interpretation of the RNASeq analysis results. TM and SH performed the GC-MS experiments for $^{13}$C labeled metabolites. MGS contributed to the design of the $^{13}$C labeled metabolic flux analysis experiment, helped with the interpretation of the results and revised the manuscript. PÇ contributed to study design and experimental planning. DM designed the study, contributed to the data analysis, drafted and revised the manuscript. All authors read and approved the final manuscript.

## Additional information

**Competing interests:** The authors declare no competing interests.

