## [Peer Review File · Nature Communications]

Reviewers' comments:

Reviewer #1 (Remarks to the Author):

Title: A single transcription factor activates the Crabtree effect in *Komagataella phaffii*

Authors: Özge Ata, Corinna Rebnegger, Nadine E. Tatto, Minoska Valli, Teresa Mairinger, Stephan Hann, Matthias G. Steiger, Pinar Çalık, Diethard Mattanovich

Summary: In their manuscript, Ata et al investigate the impact of overexpression of a Gal4-like transcription factor (termed CRA1) in the context of the respiro-fermentative phenotype in *Komagataella phaffii*. The authors demonstrate that overexpression of this single transcription factor resulted in lower specific growth rates and significantly higher glucose consumption and ethanol production. These results were then verified through analysis of the short- and long-term Crabtree phenotype and subsequent comparison to both Crabtree-positive and -negative yeasts by various metrics. This analysis enabled the authors to classify the CRA1 overexpression strain of *K. phaffii* as a Crabtree positive yeast while the control *K. phaffii* strain remained Crabtree negative. RNASeq and ¹³C labelled metabolic flux analysis were then conducted to determine the underlying expression and flux changes resulting from CRA1 overexpression.

Comments: This manuscript presents a rigorous and sound experimental study with novel and interesting findings that should be considered for publication in Nature Communications. The authors demonstrate switching *K. phaffii* from Crabtree-negative to Crabtree-positive through overexpression of a single transcription factor which is intriguing for not only understanding the global wiring of the respiro-fermentative phenotype but also for elucidating its evolutionary origins. The authors complement their experimental results and discussion with a well-founded discussion on the evolutionary mechanisms that have been proposed to understand how the Crabtree phenotype emerged and how their findings demonstrate the potential for a single mutation first evolutionary event towards respiro-fermentative metabolism. A minor comment which may help improve the manuscript is provided below:

1. Given that much of the comparison and grouping of yeast strains was conducted using experimental data from a prior report, the authors should consider testing an additional yeast strain used by Hagman et al under their experimental conditions to ensure their results with *K. phaffii* are directly comparable. While the results are sound and clearly demonstrate the switch from Crabtree-negative to Crabtree-positive, this would provide an internal control for the comparison across various yeast species from a prior report as the authors have done. The authors could also consider simply including (either in the figures or mentioned briefly in the text) the results using *P. pastoris* from Hagman et al which appear to be similar to the results with the control *K. phaffii* presented here.

Reviewer #2 (Remarks to the Author):

Authors identified the single Gal4-like transcription factor to convert Crabtree-negative *Komagataella phaffii* (*Pichia pastoris*) into a Crabtree positive characteristic. The study was well organized and well done. The results shown in the manuscript has the high impact on in the community of microbiology.

- 1) In the discussion part in page 18 to 19, authors discussed GAL gene promoters binding sites of many yeasts, and evolution of Gal4 function. Authors should describe the new TF CRA1 functions in not only *Komagataella phaffii* (*Pichia pastoris*) but also other Crabtree negative yeasts converts into Crabtree positive characteristics.

- 2) Authors should describe the Co-factor balance in the metabolism analysis. NADH should be balanced in both cases of Crabtree positive and negative characteristic. Carbon flow from pyr to ethanol should be highly related to imbalance of uptake of sugars and also close of balance of co-factors. In figure 5, authors should show the generation and consumption of co-factors.
- 3) In nature, how is this TF CRA1 regulated? What is master regulator of switch of Crabtree effect? Which environmental condition upregulate CRA1 in *Komagataella phaffii* (*Pichia pastoris*)?
- 4) Method of characteristics of transporters should be described.

Reviewer #3 (Remarks to the Author):

The authors report a functional study of strains of *K. phaffii* either deleted for the gene CRA1 or over-expressing it and compare the results with the control strain. The gene was selected based on previous results because it belongs to the GAL4 family of transcriptional activators. The main result is that over expression of CRA1 converts *K. phaffii* from a Crabtree negative yeast to a group 2 Crabtree positive yeast based on ethanol yield and biomass (Fig4). The effect on specific glucose consumption agrees with this classification but remains weak. After analyzing promoters of several genes involved in glucose or galactose utilization in several yeast species, the authors conclude that CRA1 was initially involved in glucose metabolism and only subsequently captured for galactose metabolism during evolution.

I found these results interesting and the data generally well presented, even if figures have often too small characters and symbols to read. By these criteria, the work deserves publication. My major concern is the fact that I found the arguments about evolution a little naive. Crabtree effect has been studied in some yeasts for long time and it is primarily thanks to the work of Piskur lab that it was relatively recently placed in the light of evolution using a limited number of yeast species. The loss of complex I of respiratory chain for example has occurred repetitively in the *Schizosaccharomyces pombe* which is not a budding yeast. Similarly, the role of whole-genome duplication and horizontal acquisition of URA1 are very specific to the *Saccharomyces* and related lineages, very distinct from the clade of *K. phaffii*. The conclusions of the authors on the first point are not different from a previous work, as they properly acknowledge (lines 319-320).

Figure 5 is nice but one would like to see quantitative data about actual transcript abundance, not only "unregulated" in free and "unregulated" in black. There is a main difference in this work between the extreme precision on metabolic values (do we need so precise standard deviations ?) and the global lack of precision about molecular data or strain construction (what controls were made to ensure that they are genetically correct ? and do not contain accidental spurious mutations ?).

The general conclusion that a single gene can create a Crabtree effect in yeast is valid, but the discussion about the evolution of the budding yeasts remains a bit superficial. For example, line 341 what are "evolutionary older yeasts" and then line 345 what are "more modern *Saccharomyces*"? Line 351: "the other yeast genomes contain only low numbers of binding sites in glycolytic gene promoters". I noted 16 for *Z. rouxii* and 18 for *C. albicans* (Table S3). Is this very different from 22 for *E. gossypii* or *D. hansenii* ? Compared to other figures 4, 6, 8 and 10, they look high to me.

I recommend publication after revision of the text.

Reviewer 1

Summary: In their manuscript, Ata et al investigate the impact of overexpression of a Gal4-like transcription factor (termed CRA1) in the context of the respiro-fermentative phenotype in *Komagataella phaffii*. The authors demonstrate that overexpression of this single transcription factor resulted in lower specific growth rates and significantly higher glucose consumption and ethanol production. These results were then verified through analysis of the short- and long-term Crabtree phenotype and subsequent comparison to both Crabtree-positive and -negative yeasts by various metrics. This analysis enabled the authors to classify the CRA1 overexpression strain of *K. phaffii* as a Crabtree positive yeast while the control *K. phaffii* strain remained Crabtree negative. RNASeq and ¹³C labelled metabolic flux analysis were then conducted to determine the underlying expression and flux changes resulting from CRA1 overexpression.

Comments: This manuscript presents a rigorous and sound experimental study with novel and interesting findings that should be considered for publication in Nature Communications. The authors demonstrate switching *K. phaffii* from Crabtree-negative to Crabtree-positive through overexpression of a single transcription factor which is intriguing for not only understanding the global wiring of the respiro-fermentative phenotype but also for elucidating its evolutionary origins. The authors complement their experimental results and discussion with a well-founded discussion on the evolutionary mechanisms that have been proposed to understand how the Crabtree phenotype emerged and how their findings demonstrate the potential for a single mutation first evolutionary event towards respiro-fermentative metabolism. A minor comment which may help improve the manuscript is provided below:

1. Given that much of the comparison and grouping of yeast strains was conducted using experimental data from a prior report, the authors should consider testing an additional yeast strain used by Hagman et al under their experimental conditions to ensure their results with *K. phaffii* are directly comparable. While the results are sound and clearly demonstrate the switch from Crabtree-negative to Crabtree-positive, this would provide an internal control for the comparison across various yeast species from a prior report as the authors have done. The authors could also consider simply including (either in the figures or mentioned briefly in the text) the results using *P. pastoris* from Hagman et al which appear to be similar to the results with the control *K. phaffii* presented here.

We thank the reviewer for this positive assessment, and we agree fully that it is worthwhile to place the results of Hagman et al. and our work into a directly comparable experimental context. Therefore, we have conducted further short term Crabtree effect analysis with each a strain of *S. cerevisiae* and *K. marxianus* (a Crabtree positive and a negative yeast species, which have both been studied by Hagman et al.), and we added these data as well as the *P. pastoris* data from Hagman et al. to figure 2, and explained this in the manuscript (lines 131-133). As Hagman et al. used strains of an in-house collection without providing links to their history we could not use the same strains but rather well-studied strains of each species. Our data fit well to the previously studied data, so that we think that this even broadens the scope of the overall data background on short term Crabtree effect in budding yeasts.

Reviewer #2 (Remarks to the Author):

Authors identified the single Gal4-like transcription factor to convert Crabtree-negative *Komoagataella phaffii* (*Pichia pastoris*) into a Crabtree positive characteristic. The study was well organized and well done. The results shown in the manuscript has the high impact on in the community of microbiology.

Thanks for this positive assessment.

1) In the discussion part in page 18 to 19, authors discussed GAL gene promoters binding sites of many yeasts, and evolution of Gal4 function. Authors should describe the new TF CRA1 functions in not only *Komoagataella phaffii* (*Pichia pastoris*) but also other Crabtree negative yeasts converts into Crabtree positive charactersistc.

Thank you for proposing this additional experiment. We have overexpressed the *CRA1* homolog of *Sugiyamaella lignohabitans* (another Crabtree negative yeast which we characterized in our laboratory recently – Bellasio et al. 2015, 2016). We found a 1.3-fold increase of glucose uptake rate and a 1.5-fold increase of glycerol production rate in the overexpression strains compared to the wild type. Glycerol is, besides ethanol, another reduced metabolite that serves as a redox sink in metabolic overflow conditions. These data confirm that *CRA1* also serves as a regulator of glycolysis in *S. lignohabitans*, and its overproduction leads to a metabolic overflow. These data are discussed in lines 309-315, and details are provided as supplementary table 4.

2) Authors should describe the Co-factor balance in the metabolism analysis. NADH should be balanced in both cases of Crabtree positive and negative characteristic. Carbon flow from pyr to ethanol should be highly related to imbalance of uptake of sugars and also close of balance of co-factors. In figure 5, authors should show the generation and consumption of co-factors.

Actually as correctly pointed out by the reviewer, the increase of carbon flow in glycolysis is related to the carbon flow to ethanol via the co-factor balance. In other words, the increase of specific NADH production in glycolysis of 4.7 mmol/(g*h) balances well with the increased specific NADH consumption in ethanol formation of 5.4 mmol/(g*h). We suggest to explain this in the text (lines 216-219) rather than adding even more data to figure 5 which would become rather difficult to read then.

3) In nature, how is this TF CRA1 regulated? What is master regulator of switch of Crabtree effect? Which environmental condition upregulate CRA1 in *Komoagataella phaffii* (*Pichia pastoris*)

Thank you for raising this important aspect which we have not discussed in the manuscript. There are two relevant aspects, first the comparison to *GAL4* regulation in *S. cerevisiae*, and second the transcriptional regulation of *CRA1* in *K. phaffii*. To the first aspect, *ScGAL4* is regulated at the

transcriptional level and by accessory proteins, Gal80 and Gal3. In *K. phaffii*, as well as in other Crabtree negative yeasts, there are no homologs of Gal80 and Gal3 encoded in the genome. The *KpCRA1* gene is transcriptionally upregulated on glucose surplus and on severe limitation of carbon source, when compared to limited but reasonably good supply with glucose like in the chemostat. Both aspects are discussed now in lines 293-297.

4) Method of characteristics of transporters should be described.

Has been added to the Materials and Methods section, lines 423-429.

Reviewer #3 (Remarks to the Author):

The authors report a functional study of strains of *K. phaffii* either deleted for the gene *CRA1* or over-expressing it and compare the results with the control strain. The gene was selected based on previous results because it belongs to the GAL4 family of transcriptional activators. The main result is that over expression of *CRA1* converts *K. phaffii* from a Crabtree negative yeast to a group 2 Crabtree positive yeast based on ethanol yield and biomass (Fig4). The effect on specific glucose consumption agrees with this classification but remains weak. After analyzing promoters of several genes involved in glucose or galactose utilization in several yeast species, the authors conclude that *CRA1* was initially involved in glucose metabolism and only subsequently captured for galactose metabolism during evolution.

I found these results interesting and the data generally well presented, even if figures have often too small characters and symbols to read. By these criteria, the work deserves publication.

Thank you for the positive assessment. We have revised the figures, increasing letter size and symbols where needed, and adding all primary data as requested by the journal's policy.

My major concern is the fact that I found the arguments about evolution a little naive. Crabtree effect has been studied in some yeasts for long time and it is primarily thanks to the work of Piskur lab that it was relatively recently placed in the light of evolution using a limited number of yeast species. The loss of complex I of respiratory chain for example has occurred repetitively in the *Schizosaccharomyces pombe* which is not a budding yeast. Similarly, the role of whole-genome duplication and horizontal acquisition of *URA1* are very specific to the *Saccharomyces* and related lineages, very distinct from the

clade of *K. phaffii*. The conclusions of the authors on the first point are not different from a previous work, as they properly acknowledge (lines 319-320).

Thank you for pointing this out. While we are of course aware that the Crabtree effect has evolved independently several times we did not address this properly in the manuscript. We have added further discussion on this aspect to line 256.

Figure 5 is nice but one would like to see quantitative data about actual transcript abundance, not only "unregulated" in free and "unregulated" in black. There is a main difference in this work between the extreme precision on metabolic values (do we need so precise standard deviations?) and the global lack of precision about molecular data or strain construction (what controls were made to ensure that they are genetically correct? and do not contain accidental spurious mutations?).

Concerning figure 5 it was our intention not to overload it by too many data around the single reactions, and therefore provided the regulation data (fold changes) to the supplemental data. However we agree that it is worth showing them in the main body of the paper, and added them as an inserted table to figure 5. We hope that you agree that the precision of these data matches the metabolic data. Concerning the strain construction, we fully agree that accidental additional mutations are a relevant concern, which is why we repeated the overexpression strain construction several times during this work. All shake flask and chemostat cultures were run in parallel with independent transformants, and transformation was repeated in different strain backgrounds as well, all leading to the same phenotype. Therefore we conclude that the observed phenotype is caused by *CRA1* overexpression and not by an adventitious mutation. This aspect on strain construction was added now to Materials and Methods, lines 331-337.

The general conclusion that a single gene can create a Crabtree effect in yeast is valid, but the discussion about the evolution of the budding yeasts remains a bit superficial. For example, line 341 what are "evolutionary older yeasts" and then line 345 what are "more modern *Saccharomyces*"? Line 351: "the other yeast genomes contain only low numbers of binding sites in glycolytic gene promoters". I noted 16 for *Z. rouxii* and 18 for *C. albicans* (Table S3). Is this very different from 22 for *E. gossypii* or *D. hansenii*? Compared to other figures 4, 6, 8 and 10, they look high to me. I recommend publication after revision of the text.

The language of the text was revised, and we have corrected the manuscript to ensure precise expressions in the discussion of evolution. Concerning the numbers in table S3, the reviewer is absolutely right that we forgot to mention *C. albicans* with Gal4 binding sites in glycolytic but not in galactolytic gene promoters. *Z. rouxii* seems to be a transitional form with Gal4 binding sites related to

both pathways (such as *K. naganishii*). This is revised and properly noted now in lines 302-305 of the manuscript.

REVIEWERS' COMMENTS:

Reviewer #1 (Remarks to the Author):

The authors have satisfactorily addressed my comments to the original version of their manuscript. Specifically, they conducted additional experiments to characterize the short term Crabtree effect in strains of *S. cerevisiae* and *K. marxianus* and included the results in the revised manuscript.

Reviewer #2 (Remarks to the Author):

The manuscript was revised well and authors answered to reviewer's comments and questions accordingly.

Reviewer #3 (Remarks to the Author):

This is an improved version of the manuscript that answers all my previous questions and takes into account my remarks. I suggest to accept it.

REVIEWERS' COMMENTS:

Reviewer #1 (Remarks to the Author):

The authors have satisfactorily addressed my comments to the original version of their manuscript. Specifically, they conducted additional experiments to characterize the short term Crabtree effect in strains of *S. cerevisiae* and *K. marxianus* and included the results in the revised manuscript.

Reviewer #2 (Remarks to the Author):

The manuscript was revised well and authors answered to reviewer's comments and questions accordingly.

Reviewer #3 (Remarks to the Author):

This is an improved version of the manuscript that answers all my previous questions and takes into account my remarks. I suggest to accept it.